# Bridging SFT and DPO for Diffusion Model Alignment with Self-Sampling Preference Optimization

## Abstract

Existing post-training techniques are broadly categorized into supervised fine-tuning (SFT) and reinforcement learning (RL) methods; the former is stable during training but suffers from limited generalization, while the latter, despite its stronger generalization capability, relies on additional preference data or reward models and carries the risk of reward exploitation. In order to preserve the advantages of both SFT and RL – namely, eliminating the need for paired data and reward models while retaining the training stability of SFT and the generalization ability of RL – a new alignment method, Self-Sampling Preference Optimization (SSPO), is proposed in this paper. SSPO introduces a Random Checkpoint Replay (RCR) strategy that utilizes historical checkpoints to construct paired data, thereby effectively mitigating overfitting. Simultaneously, a Self-Sampling Regularization (SSR) strategy is employed to dynamically evaluate the quality of generated samples; when the generated samples are more likely to be winning samples, the approach automatically switches from DPO (Direct Preference Optimization) to SFT, ensuring that the training process accurately reflects the quality of the samples. Experimental results demonstrate that SSPO not only outperforms existing methods on text-to-image benchmarks, but its effectiveness has also been validated in text-to-video tasks. We validate SSPO across both text-to-image and text-to-video benchmarks. SSPO surpasses all previous approaches on the text-to-image benchmarks and demonstrates outstanding performance on the text-to-video benchmarks.

## 1 Introduction

Text-to-visual models (Huang et al., 2025; Li et al., 2024c) have become a crucial component of the AIGC (AI-generated content) industry, with the denoising diffusion probabilistic model (DDPM) (Ho et al., 2020; Kingma et al., 2021) being the most widely used technology. However, current pre-trained text-to-visual models often fail to adequately align with human requirements. As a result, post-training techniques (Liang et al., 2024a; Black et al., 2024; Wallace et al., 2024; Yang et al., 2024; Li et al., 2024a;b) are widely used to align pre-trained models to better satisfy human needs.

Currently, post-training techniques that are widely used can generally be categorized into two main types: supervised fine-tuning (SFT) and reinforcement learning (RL) methods. However, each approach has its advantages and limitations. SFT requires only target distribution data for training and exhibits greater stability in the training process compared to RL. However, its generalization ability is significantly inferior to that of RL methods (Chu et al., 2025). Conversely, common RL methods, despite their strong generalization capabilities, often necessitate additional data or model constraints. For instance, Diffusion-DPO (Wallace et al., 2024) employs a fixed set of preference data (generated by humans or other models) as training data. Methods such as DDPO (Black et al., 2024), CDPO (Croitoru et al., 2024), and SPO (Liang et al., 2024b) utilize a reward model (RM) (Escontrela et al., 2024) to score outputs and subsequently backpropagate policy gradients based on the scoring results. This not only incurs additional computational costs but also introduces the risk of reward hacking (Denison et al., 2024). Moreover, in the context of text-to-visual tasks, despite the existence of numerous evaluation models, finding a solution capable of providing comprehensive feedback on all aspects of visual content remains challenging (Kim et al., 2024). Also, constructing an effective and efficient reward model is particularly difficult, as it heavily relies on the collection of costly

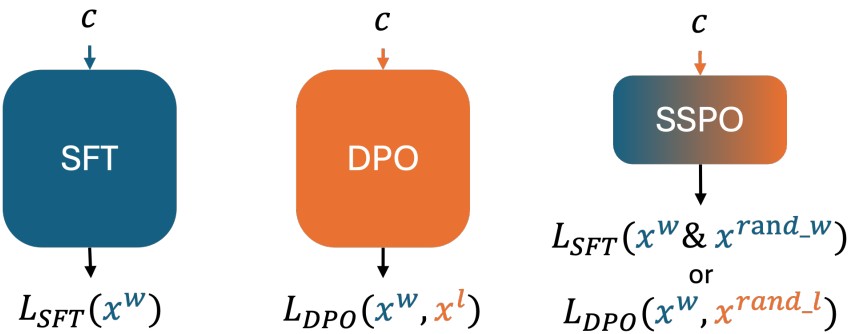

Figure 1: Illustration of SFT, DPO, and the proposed SSPO. $x^w$ is the winning samples, and $x^l$ is the losing samples. $x^{\mathrm{rand\_w}}$ and $x^{\mathrm{rand\_l}}$ are the samples generated by randomly sampled checkpoints. SSPO can switch between SFT and DPO based on the current state of the datapoint.

paired feedback data. This naturally raises the question of whether it is possible to develop an approach that combines the advantages of both SFT and RL – eliminating the need for paired preference data or reward models required by RL methods, while retaining the training stability of SFT and achieving a level of generalization comparable to RL. In this work, as shown in Figure. 1, we introduce a method called Self-Sampling Preference Optimization (SSPO) to tackle this challenge.

Since only SFT data ("winning" samples) can be used, we need to construct appropriate new samples to complete the DPO training process. Some previous methods have utilized sampling from historical distributions to construct new samples, such as Experience Replay (ER) Lin (1992); Schmitt et al. (2018); Pinto et al. (2017). ER improves learning efficiency and effectiveness by storing and reusing past experiences (historical data stored in the buffer). This concept is also reflected in SPIN-Diffusion Yuan et al. (2024a), where samples are drawn from the latest checkpoint to generate negative samples. However, such methods face two major challenges:

- *What kind of historical distribution is worth sampling?*

- *Are the sampled data truly negative samples?*

To address the first issue, we employ historical checkpoints as the minimal sampling unit of the Experience Replay Distribution (ERD). Specifically, we conduct experiments using three different approaches to identify the optimal sampling strategy for ERD: (1) consistently selecting the initial checkpoint as the ERD, (2) selecting the last checkpoint saved from the previous iteration, and (3) uniform sampling from all previous checkpoints. Our findings indicate that as the number of training steps increases, compared to always utilizing the last checkpoint, selecting the checkpoint from the previous iteration can mitigate the risk of overfitting and yields superior performance. Furthermore, randomly selecting from all previous checkpoints produces comparable results to selecting the checkpoint from the previous iteration while contributing to a more stable overall training process. Therefore, we refer to this final strategy as **Random Checkpoint Replay (RCR)**, which enables the model to learn from historical information better, ensuring both training stability and outstanding performance.

Meanwhile, regarding the second issue, previous strategies lack any guarantee that the sampled data points are necessarily negative samples. However, in DPO, a well-defined "winning" or "losing" paired dataset is essential. Therefore, to determine whether the data generated by the reference model are genuinely "losing" or "winning" samples relative to the current policy, we propose a novel strategy called **Self-Sampling Regularization (SSR)**. This strategy enables the classification of sampled examples as either positive or negative samples. Specifically, if a winning data point is available for the model to learn from and the reference model exhibits a higher probability than the current model of generating this winning data point, then the probability of sampling a winning data point from the reference modelâĂŹs distribution will exceed that of generating a losing data point. In other words, under such conditions, the so-called "losing" samples

do not constitute truly negative samples for the current model. In such cases, SSPO automatically shifts the learning paradigm from DPO to SFT, thereby preventing the model from learning biases from samples that are not genuinely "losing" when using the DPO algorithm.

In summary, the main contributions of this work are:

1. We design a new preference algorithm SSPO that, for the first time, integrates the advantages of SFT and DPO, achieving superior optimization results.

2. We design the Random Checkpoint Replay strategy to construct paired data, then experimentally and theoretically analyze it as an effective solution in sampling of the experience replay distribution.

3. We further design a criterion termed Self-sampling Regularization to evaluate the quality of the generated responses, which subsequently determines whether the current sample is used for DPO or SFT.

4. SSPO outperforms all previous optimization methods on the text-to-image benchmarks, and is also effective on the text-to-video benchmarks.

## 2 Related Work

**Self-Improvement.** Self-improvement methods use iterative sampling to improve pre-trained models, which has the potential to eliminate the need for an expert annotator such as a human or more advanced models. A recent work INPO (Zhang et al., 2025) directly approximates the Nash policy (the optimal solution) of a two-player game on a human-annotated preference dataset. Concurrently, the demonstrated self-improvement methods (Shaikh et al., 2024; Chen et al., 2024) align pre-trained models only using supervised learning dataset (one demonstrated response for each prompt) without using reward models and human-annotated data pairs. Specifically, it takes the demonstrated response as the winner and generates the loser by reference models. DITTO (Shaikh et al., 2024) fixes the reference model as the initial model and works well on small datasets. SPIN (Chen et al., 2024) takes the latest checkpoint as the reference model and uses it to generate responses in each iteration. However, these approaches have the following shortcomings. First, they only focus on transformer-based models for the text modality. Second, the selection of reference models is fixed, which has limited space for policy exploration. Third, either a human-annotated preference dataset is necessarily required or they are built on a strong assumption that the demonstrated responses are always preferred to the generated responses. In fact, the responses generated by the models are not necessarily bad and do not always need to be rejected.

**RM-Free PO for Diffusion Model Alignment.** DPO-style methods (Rafailov et al., 2024) use a closed-form expression between RMs and the optimal policy to align pre-trained models with human preference. Thus, no RM is needed during the training process. Diffusion-DPO is first proposed in Wallace et al. (2024). Based on offline datasets with labeled human preference, it shows promising results on text-to-image diffusion generation tasks. Diffusion-RPO (Gu et al., 2024) considers semantic relationships between responses for contrastive weighting. MaPO (Hong et al., 2024) uses margin-aware information, and removes the reference model. Diffusion-NPO (Wang et al., 2025) further adds label smoothing on Diffusion-DPO. However, only off-policy data is considered and human-annotated data pairs are necessarily required with high labor costs.

To involve on-policy data and minimize human annotation costs, self-improvement methods are being explored. SPIN-Diffusion (Yuan et al., 2024a) adapts the SPIN method to text-to-image generation tasks with diffusion models. It shows high efficiency in data utilization. However, first, it has the issues mentioned above as a self-improvement method. Second, only the text-to-image generation task is considered in all previous works.

## 3 Background

### 3.1 Denoising Diffusion Probabilistic Model

Given a data sample $\mathbf{x}_0 \sim \mathcal{D}$, the forward process is a Markov chain that gradually adds Gaussian noise to the sample as follows

$$q(\mathbf{x}_{1:T}|\mathbf{x}_0) = \prod_{t=1}^{T} q(\mathbf{x}_t|\mathbf{x}_{t-1}), \quad q(\mathbf{x}_t|\mathbf{x}_{t-1}) = \mathcal{N}(\mathbf{x}_t|\sqrt{\alpha}\mathbf{x}_{t-1}, (1-\alpha)\mathbf{I}), \tag{1}$$

where $\mathbf{x}_1, \cdots, \mathbf{x}_T$ are latent variables, and $\alpha$ is a noise scheduling factor. Equivalently, $\mathbf{x}_t = \sqrt{\alpha}\mathbf{x}_{t-1} + \sqrt{1-\alpha}\epsilon$, where $\epsilon \sim \mathcal{N}(0, \mathbf{I})$.

With a latent variable $\mathbf{x}_t$ from the forward process, DDPM estimates the normalized additive noise by $\epsilon_\theta(\mathbf{x}_t)$, where $\theta$ represents the parameters of the neural network. To *maximize* the evidence lower bound (ELBO) (Kingma & Welling, 2019), we usually *minimize* the loss function w.r.t. $\theta$:

$$\mathcal{L}_{\text{DM}}(\theta; \mathbf{x}_0) = \mathbb{E}_{\epsilon, t}\left[w_t \|\epsilon_\theta(\mathbf{x}_t) - \epsilon\|^2\right], \tag{2}$$

where $t \sim \mathcal{U}(1, T)$ is uniformly distributed on the integer interval $[1, T]$, $w_t = \frac{(1-\alpha)^2 \alpha^{t-1}}{2\sigma_t^2 (1-\alpha^t)^2}$ is a weighting scalar, and $\sigma_t^2 = \frac{(1-\alpha)(1-\alpha^{\frac{t-1}{2}})}{1-\alpha^t}$ is the variance of the additive noise. $T$ as a pre-specified constant in Kingma et al. (2021) is ignored in the loss function, because it has no contribution to training. In practice, $w_t$ is usually set to a constant (Song & Ermon, 2019).

### 3.2 Direct Preference Optimization

Given a prompt or condition $\mathbf{c}$, the human annotates the preference between two results as $\mathbf{x}^w \succ \mathbf{x}^l$. After preference data collection, in conventional RLHF (Reinforcement Learning from Human Feedback), we train a reward function based on the Bradley-Terry (BT) model. The goal is to maximize cumulative rewards with a Kullback–Leibler (KL) constraint between the current model $\pi_\theta$ and a reference model $\pi_{\text{ref}}$ (usually the initial model) as follows

$$\max_{\pi_\theta} \mathbb{E}_{\substack{\mathbf{c} \sim \mathcal{D}, \\ \mathbf{x} \sim \pi_\theta(\cdot|\mathbf{c})}} \left[ r(\mathbf{x}, \mathbf{c}) - \beta D_{\text{KL}}\big(\pi_\theta(\mathbf{x}|\mathbf{c}) || \pi_{\text{ref}}(\mathbf{x}|\mathbf{c})\big) \right], \tag{3}$$

where $r(\cdot)$ is the reward function induced from the BT model, and $\beta$ is a hyperparameter to control the weight of the KL constraint.

In DPO, the training process is simplified, and the target is converted to minimize a loss function as follows

$$\mathcal{L}_{\text{DPO}}(\theta) = \mathbb{E}_{(\mathbf{x}^w, \mathbf{x}^l, \mathbf{c}) \sim \mathcal{D}} \left[ -\log \sigma\left(\beta \log \frac{\pi_\theta(\mathbf{x}^w|\mathbf{c})}{\pi_{\text{ref}}(\mathbf{x}^w|\mathbf{c})} - \beta \log \frac{\pi_\theta(\mathbf{x}^l|\mathbf{c})}{\pi_{\text{ref}}(\mathbf{x}^l|\mathbf{c})}\right) \right], \tag{4}$$

where $\sigma(\cdot)$ (without subscript) is the logistic function.

## 4 Self-Sampling Preference Optimization

### 4.1 Random Checkpoint Replay

When we only have supervised fine-tuning data $(\mathbf{c}, \mathbf{x}^w)$, curating the paired data $\mathbf{x}^{\text{rand}}$ becomes crucial. However, how to select an appropriate ERD remains a challenge. Moreover, unlike DPO, SSPO combines both SFT and DPO, which makes the reference model more influenced by the ERD rather than the initial model. Consequently, we adopt the ERD as our reference model. To further investigate the impact of ERD sampling, we design and evaluate three sampling strategies for the ERD as follows:

- The ERD is sampled from the initial model, and is denoted as ERD = 0.

- The ERD is sampled from the checkpoint saved in the last iteration, and is denoted as $\text{ERD} = k - 1$.

- The ERD is uniformly sampled from all previously saved checkpoints, and is denoted as $\text{ERD} = [0, k - 1]$.

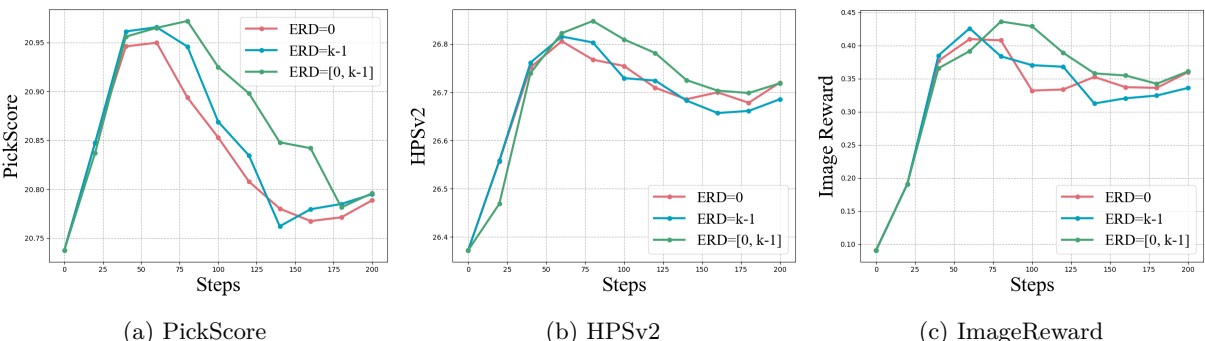

(a) PickScore            (b) HPSv2            (c) ImageReward

Figure 2: **Ablation study of the Experience Replay Distribution selection strategies.** The x-axis is the number of training steps $T$. The y-axis is the testing score. (The higher is better.)

We then use ERD to generate the pseudo "losing samples" required for the DPO training process. Thus, in each iteration, after omitting $\mathbf{c}$ for the sake of concision, the loss function adapted from diffusion DPO is as follows

$$
\begin{aligned}
\mathcal{L}(\theta; \mathbf{x}_0^w, \mathbf{x}_0^{\text{rand}}) \; = \; & \mathbb{E}_{\epsilon^w, \epsilon^{\text{ref}}, t} \Big[ - \log \cdot \\
& \sigma\Big( - \beta T w_t (\|\epsilon_\theta(\mathbf{x}_t^w) - \epsilon^w\|^2 - \|\epsilon_{\text{ref}}(\mathbf{x}_t^w) - \epsilon^w\|^2 - \|\epsilon_\theta(\mathbf{x}_t^{\text{rand}}) - \epsilon^{\text{ref}}\|^2 + \|\epsilon_{\text{ref}}(\mathbf{x}_t^{\text{rand}}) - \epsilon^{\text{ref}}\|^2) \Big) \Big],
\end{aligned}
\tag{5}
$$

where $t \sim \mathcal{U}(1, T)$, $\epsilon^w, \epsilon^{\text{ref}} \overset{\text{iid}}{\sim} \mathcal{N}(0, \mathbf{I})$, and $\mathbf{x}_t^{w, \text{rand}} = \sqrt{\alpha^t} \mathbf{x}_0^{w, \text{rand}} + \sqrt{1 - \alpha^t} \epsilon^{w, \text{ref}}$. Notably, in equation 5, the reference image $\mathbf{x}_0^{\text{rand}}$ is generated from the ERD distribution, which is on-policy learning in the last two settings.

We use the Pick-a-Pic dataset and the stable diffusion 1.5 (SD-1.5) model to conduct all experiments for this study. In experiments, we save a checkpoint every 30 updates ($K = 7$ iterations). The experimental results are shown in Figure 2. In the beginning, the performances of all models are improved. However, as the number of steps $T$ increases, the performance of the model with $\text{ERD} = 0$ starts to decrease and quickly exhibits overfitting. The model with $\text{ERD} = k - 1$ shows unstable performances. Due to the fact that when all "losing samples" are generated from the last checkpoint in the current round of training, the model is prone to fall into non-global optima, leading to instability in the training process Florensa et al. (2018). Therefore, considering both model performance and stability, our method uses $\text{ERD} = [0, k - 1]$ to generate reference samples, referring to this approach as Random Checkpoint Replay.

Specifically, in Theorem 4.1, under the PAC-Bayesian framework Germain et al. (2016); Seeger (2002), we theoretically show that our uniformly sampling strategy has better generalization ability compared to the latest checkpoint selection strategy, e.g., SPIN.

**Theorem 4.1** *With uniformly sampling from previous checkpoints $[0, K - 1]$, the PAC-Bayesian upper bound of the generalization loss is guaranteed to be smaller or equal to the latest checkpoint.*

$$
\mathcal{O}\Big( \underset{\pi_\theta \sim Q_{\text{UNI}}}{\mathbb{E}} \big[ L(\pi_\theta) \big] \Big) \leq \mathcal{O}\Big( \underset{\pi_\theta \sim Q_K}{\mathbb{E}} \big[ L(\pi_\theta) \big] \Big),
\tag{6}
$$

*where $L(\pi_\theta) \coloneqq \mathbb{E}_{\mathbf{x} \sim \mathcal{D}} \mathcal{L}(\theta; \mathbf{x})$ is the generalization loss on the policy, $Q_{\text{UNI}}$ is the posterior policy distribution with uniformly sampling, and $Q_K$ is the posterior distribution with the latest checkpoint.*

The complete proof is given in Appendix A.1.

## 4.2 Self-Sampling Regularization

On the other hand, is the data sampled from ERD truly "losing samples" relative to the current policy? Obviously, when generating the samples $\mathbf{x}_0^{\text{rand}}$, we cannot determine the relative relationship between the ERD and the current model (whether the image sampled from ERD is better than the image generated by the current model or not). Existing methods always directly assume that the ERD samples have relatively poor quality. For example, DITTO assumes that the earlier checkpoints used as the reference model produce worse-performing samples. However, as shown in the experiment in Figure 2, the ERD is not always inferior to the current model.

Specifically, based on the DPO-style optimization method, we analyze the change in gradient updates when the ERD image is **better** than the image generated by the current model. The gradient of the loss function in equation 5 is as follows

$$
\nabla \mathcal{L}(\theta; \mathbf{x}_0^w, \mathbf{x}_0^{\text{rand}}) = \underset{\epsilon^w, \epsilon^{\text{ref}}, t}{\mathbb{E}} \left[ -2\beta T w_t \sigma \left( -\beta T w_t (\hat{\sigma}_{\text{ref}}^2 - \hat{\sigma}_w^2) \right) \underbrace{\left( \epsilon_\theta(\mathbf{x}_t^w) - \epsilon^w - \epsilon_\theta(\mathbf{x}_t^{\text{rand}}) + \epsilon^{\text{ref}} \right)}_{\text{error term}} \right], \tag{7}
$$

where $\hat{\sigma}_w^2 := \|\epsilon_\theta(\mathbf{x}_t^w) - \epsilon^w\|^2 - \|\epsilon_{\text{ref}}(\mathbf{x}_t^w) - \epsilon^w\|^2$ and $\hat{\sigma}_{\text{ref}}^2 := \|\epsilon_\theta(\mathbf{x}_t^{\text{rand}}) - \epsilon^{\text{ref}}\|^2 - \|\epsilon_{\text{ref}}(\mathbf{x}_t^{\text{rand}}) - \epsilon^{\text{ref}}\|^2$. If $\mathbf{x}_t^{\text{rand}}$ generated by the ERD is good and close to $\mathbf{x}_t^w$, both the error term and the weight term $\sigma(\cdot)$ tend to be small. This leads to a very small gradient update without making full use of the information in the data sample, even though we know that $\mathbf{x}_0^w$ is generated by experts with high quality.

Therefore, in order to avoid the impact of the uncertainty of the image sampled from ERD, we design a strategy that uses the winning image as an anchor, and evaluates the quality of the ERD samples based on its performance relative to the current model on the anchor. Specifically, we design the Self-sampling Regularization (SSR) to the loss function in equation 5 as follows:

$$
\mathcal{L}(\theta; \mathbf{x}_0^w, \mathbf{x}_0^{\text{rand}}) = \underset{\epsilon^w, \epsilon^{\text{ref}}, t}{\mathbb{E}} \left[ -\log \cdot \right.
$$
$$
\left. \sigma \left( -\beta T w_t \left( \|\epsilon_\theta(\mathbf{x}_t^w) - \epsilon^w\|^2 - \|\epsilon_{\text{ref}}(\mathbf{x}_t^w) - \epsilon^w\|^2 - \text{sign} \cdot (\|\epsilon_\theta(\mathbf{x}_t^{\text{rand}}) - \epsilon^{\text{ref}}\|^2 - \|\epsilon_{\text{ref}}(\mathbf{x}_t^{\text{rand}}) - \epsilon^{\text{ref}}\|^2) \right) \right) \right], \tag{8}
$$

where $\epsilon^w, \epsilon^{\text{ref}} \overset{\text{iid}}{\sim} \mathcal{N}(0, \mathbf{I})$ and $\mathbf{x}_t^{w, \text{rand}} = \sqrt{\alpha^t} \mathbf{x}_0^{w, \text{rand}} + \sqrt{1 - \alpha^t} \epsilon^{w, \text{ref}}$. sign is a binary variable defined as

$$
\text{sign} = \text{sgn}(\|\epsilon_{\text{ref}}(\mathbf{x}_t^w) - \epsilon^w\|^2 - \|\epsilon_\theta(\mathbf{x}_t^w) - \epsilon^w\|^2), \tag{9}
$$

where $\text{sgn}(x) = 1$ if $x > 0$, and otherwise $\text{sgn}(x) = -1$.

**Bridging SFT and DPO.** We give the pseudocode of SPPO in Algorithm 1. The integration of our random checkpoint replay and self-sampling regularization strategies facilitates seamless switching between SFT and DPO, by leveraging results from randomly sampled checkpoints. Specifically in Line 4 of Algorithm 1, the loss function is calculated using equation 8, where function sign judges whether SFT or DPO is performed. If $\text{sign} = 1$, it performs DPO updates with $\mathbf{x}_0^w$ and $\mathbf{x}_0^{\text{rand}_w}$ targets. If $\text{sign} = -1$, the performance is equivalent to SFT updates with $\mathbf{x}_0^w$ and $\mathbf{x}_0^{\text{rand}_l}$ labels, because the $\log \sigma(\cdot)$ function is monotonic. We validate the effectiveness of SSR through the ablation study in Table 6 in Section 5.2.4 and show the SSR Rate in Figure 3 where the definition of SSR Rate is given in Section 5.2.1.

Intuitively, if the sampled checkpoint has a higher probability of generating noise $\epsilon^w$ compared to the current model, then in this situation, the reference image generated by the sampled checkpoint is also more likely to be a winning image than a losing image. We formalize this claim in Theorem 4.2, which motivates the design of the criterion in equation 9.

**Theorem 4.2** *Given two policy model parameters $\theta_1$ and $\theta_2$, it is almost certain that $\theta_1$ generates better prediction than $\theta_2$, if*

$$
\mathbb{E} \left[ \|\epsilon_{\theta_1}(\mathbf{x}_t) - \epsilon\|^2 - \|\epsilon_{\theta_2}(\mathbf{x}_t) - \epsilon\|^2 \right] \leq 0, \tag{10}
$$

*where $\epsilon \sim \mathcal{N}(0, \mathbf{I})$, and $\mathbf{x}_t = \sqrt{\alpha^t} \mathbf{x}_0 + \sqrt{1 - \alpha^t} \epsilon$.*

---

**Algorithm 1** SSPO

---

**Require:** Demonstrated data set $(\mathbf{x}_0^w, \mathbf{c}) \sim \mathcal{D}$; Number of diffusion steps $T$; Number of iterations $K$; Initial model $\theta_0$.
1: **for** $k = 1, \cdots, K$ **do**
2:     Sample a checkpoint rand $\sim \mathcal{U}(0, k - 1)$.
3:     Generate $\mathbf{x}_0^{\text{rand}}$ from $\theta_{\text{rand}}$, and compose data pairs $(\mathbf{x}_0^w, \mathbf{x}_0^{\text{rand}}, \mathbf{c})$.
4:     Compute $\mathcal{L}(\theta; \mathbf{x}_0^w, \mathbf{x}_0^{\text{rand}})$ according to equation 8.
5:     $\theta_k \leftarrow \theta_{k-1} - \eta_{k-1} \nabla \mathcal{L}(\theta; \mathbf{x}_0^w, \mathbf{x}_0^{\text{rand}})$     # Or other optimizer, e.g., AdamW.
6: **end for**
**Ensure:** $\theta_K$

---

The proof of Theorem 4.2 is given in Appendix A.2. According to Theorem 4.2, if sign $= -1$, it means that $\mathbf{x}_0^{\text{rand}}$ generated by $\epsilon_{\text{ref}}$ has a high probability to be better than the output of $\epsilon_\theta$. In this situation, $\mathbf{x}_0^{\text{rand}}$ is of good quality and should not be rejected. The effectiveness of the SSR is further verified in the ablation study in Section 5.

## 5 Experiments

### 5.1 Setup

#### 5.1.1 Text-to-Image

In the text-to-image task, we test our methods based on the stable diffusion 1.5 (SD-1.5) model. Following Diffusion-DPO (Wallace et al., 2024), we use the sampled training set of the Pick-a-Pic v2 dataset (Kirstain et al., 2023) as the training dataset. Pick-a-Pic dataset is a human-annotated preference dataset for image generation. It consists of images generated by the SDXL-beta (Podell et al., 2024) and Dreamlike models. Specifically, as mentioned in Diffusion-DPO, we remove approximately 12% pairs with ties and use the remaining $851, 293$ pairs, which include $58, 960$ unique prompts for training. We use AdamW (Loshchilov & Hutter, 2019) as the optimizer. We train our model on 8 NVIDIA A100 GPUs with local batch size 1, and the number of gradient accumulation steps is set to 256. Thus, the equivalent batch size is 2048. A learning rate of $5 \times 10^{-9}$ is used, as we find that a smaller learning rate can help avoid overfitting. We set $\beta$ to 2000, which stays the same in Diffusion-DPO. For evaluation datasets, we use the validation set of the Pick-a-Pic dataset, the Parti-prompt, and HPSv2, respectively. We utilize the default stable diffusion inference pipeline from Huggingface when testing. The metrics we use are PickScore, HPSv2, ImageReward, and Aesthetic score.

#### 5.1.2 Text-to-Video

To further verify that SSPO works well in text-to-video generation tasks, we test our methods based on the AnimateDiff (Guo et al., 2024). We use the training set from MagicTime (Yuan et al., 2024b) as our training set and utilize the ChronoMagic-Bench-150 (Yuan et al., 2024c) dataset as our validation set. We use LoRA (Hu et al., 2022) to train all the models at the resolution $256 \times 256$ and 16 frames are taken by dynamic frames extraction from each video. The learning rate is set to $5 \times 10^{-6}$ and the training steps are set to 1000. The metrics we use are FID (Heusel et al., 2017), PSNR (Hore & Ziou, 2010) and FVD (Unterthiner et al., 2019), MTScore (Yuan et al., 2024c) and CHScore(Yuan et al., 2024c).

### 5.2 Results

#### 5.2.1 Analysis of Text-to-Image

To verify the effectiveness of the proposed SSPO, we compare SSPO with the SOTA methods, including DDPO (Black et al., 2024), DPOK (Fan et al., 2023), D3PO (Yang et al., 2024), Diffusion-DPO, Diffusion-RPO (Gu et al., 2024), SPO (Liang et al., 2024b) and SPIN-Diffusion. We first report all the comparison results on the validation unique split of Pick-a-Pic dataset in Table 1. SSPO outperforms previous methods

across most metrics, even those that utilize reward models during the training process, such as DDPO and SPO. Moreover, SSPO does not require the three-stage training process like SPIN-Diffusion, nor does it require the complex selection of hyperparameters. Notably, we observe that SSPO significantly improves ImageReward, which may be attributed to the fact that ImageReward reflects not only the alignment between the image and human preference but also the degree of alignment between the image and the text. In contrast, the other metrics primarily reflect the alignment between the image and human preference.

Previous methods like SPIN-Diffusion used the winning images from the Pick-a-Pic dataset as training data. In our experiments, aside from training using the winning images as done previously, we also conduct an experiment where we randomly select images from both the winner and the losing sets as training data for SSPO, which we refer to as $SSPO^r$. Despite using a lower-quality training data distribution, $SSPO^r$ still outperforms other methods on most metrics. Notably, when compared to SFT, which is fine-tuned on the winning images, $SSPO^r$ still exceeds SFT on three key metrics, further demonstrating the superiority of SSPO.

To better evaluate SSPO's out-of-distribution performance, we also test the model using the HPSv2 and Parti-prompt datasets, which have different distributions from the Pick-a-pic dataset. As shown in Table 2 and Table 7 in the Appendix, SSPO outperforms all other models on these datasets. It is worth noting that SPO has not been tested on these two datasets and SPIN-Diffusion does not report their precise results. So, we reproduce the results by using the checkpoints of SPIN-Diffusion[1] and SPO[2] available on HuggingFace.

We also evaluate SSPO on the Pick-a-Pic validation set using the SDXL backbone to assess its performance in a more advanced generation setting. As shown in Table 3, SSPO consistently outperforms all baseline methods on PickScore, HPSv2, and Image Reward, indicating stronger alignment with human preference and learned reward signals. Although SPO achieves a slightly higher Aesthetic score, SSPO demonstrates superior overall performance across key metrics. These results further validate the effectiveness of SSPO in enhancing model feedback quality, even when applied to stronger base models like SDXL.

In Figure 3, we further explore the relationship between our SSR rate and model performance. SSR rate is defined as the ratio $(\#sign = -1)/(\#total)$ in a batch of data. The SSR rate reflects the comparative performance between the current model and all previous checkpoints; A higher SSR rate indicates that the samples generated by the ERD are more likely to be positive samples compared to the current model, meaning that the current model is competitive with ERD which relative to all previous models. We observe that in the early stages of training, a low SSR rate enables the model to quickly learn information from positive samples. In the later stages, as the SSR rate increases, the current model no longer holds an absolute advantage, indicating that the model has effectively learned the distribution of the sample data.

We also visualize the results of SD-1.5, SPO, SPIN-Diffusion, and SSPO in Figure 5. SSPO is able to capture the verb "nested" and also generates better eye details. SSPO not only demonstrates superior visual quality compared to other methods but also excels in image-text alignment. This is because SSPO's training approach, which does not rely on RMs to guide the learning direction, allows the model to learn both human preferences and improve in areas that RMs may fail to address. We further conduct human evaluation experiments, with the experimental details and results provided in the Appendix B.3.

### 5.2.2 Memory and Time Cost

We show the comparison of GPU and Time cost as well as performance comparisons in Table 4. We set the batch size to 1 and calculated the average time cost and maximum GPU usage over 100 data points for each method. SSPO is comparable to SPIN, DPOK, etc., in terms of GPU usage and execution time, second only to DPO. This is because our method does not require any reward model, eliminating the need for GPU usage and inference time of a reward model. Compared to other methods, our approach also requires fewer hyperparameters. In SSPO, compared to Diffusion-DPO, the only hyperparameter introduced is the number of iterations $K$ where we study the ablation in Figure. 4.

---

[1] https://huggingface.co/UCLA-AGI/SPIN-Diffusion-iter3
[2] https://huggingface.co/SPO-Diffusion-Models/SPO-SD-v1-5_4k-p_10ep

Table 1: **Model Feedback Results on the Pick-a-Pic Validation Set.** SSPO$^r$ indicate that we use **randomly chosen** images in the Pick-a-Pic dataset as the training set. IR indicates Image Reward.

| Methods | PickScore ↑ | HPSv2 ↑ | IR ↑ | Aesthetic ↑ |
|---------|-------------|---------|------|-------------|
| SD-1.5 | 20.53 | 23.79 | -0.163 | 5.365 |
| SFT | 21.32 | 27.14 | 0.519 | 5.642 |
| DDPO | 21.06 | 24.91 | 0.082 | 5.591 |
| DPOK | 20.78 | 24.08 | -0.064 | 5.568 |
| D3PO | 20.76 | 23.91 | -0.124 | 5.527 |
| Diff-DPO | 20.98 | 25.05 | 0.112 | 5.505 |
| SPO | 21.43 | 26.45 | 0.171 | 5.887 |
| SPIN-Diff | 21.55 | 27.10 | 0.484 | **5.929** |
| SSPO$^r$ | 21.33 | 27.07 | 0.524 | 5.712 |
| SSPO | **21.57** | **27.20** | **0.615** | 5.772 |

Table 2: **Model Feedback Results on the HPSv2 Dataset.** IR indicates Image Reward.

| Methods | PickScore ↑ | HPSv2 ↑ | IR ↑ | Aesthetic ↑ |
|---------|-------------|---------|------|-------------|
| SD-1.5 | 20.95 | 27.17 | 0.08 | 5.55 |
| SFT | 20.95 | 27.88 | 0.68 | 5.82 |
| Diff-DPO | 20.95 | 27.23 | 0.30 | 5.68 |
| Diff-RPO | 21.43 | 27.37 | 0.34 | 5.69 |
| SPO | 21.87 | 27.60 | 0.41 | 5.87 |
| SPIN-Diff | 21.88 | 27.71 | 0.54 | **6.05** |
| SSPO | **21.90** | **27.92** | **0.70** | 5.94 |

Table 3: **Model Comparisons on SDXL on the Pick-a-Pic Validation Set.** IR indicates Image Reward.

| Methods | PickScore ↑ | HPSv2 ↑ | IR ↑ | Aesthetic ↑ |
|---------|-------------|---------|------|-------------|
| SDXL | 21.95 | 26.95 | 0.54 | 5.95 |
| Diff-DPO | 22.64 | 29.31 | 0.94 | 6.02 |
| MAPO | 22.11 | 28.22 | 0.72 | 6.10 |
| SPO | 23.06 | 31.80 | 1.08 | **6.36** |
| SSPO | **23.24** | **32.04** | **1.16** | 6.32 |

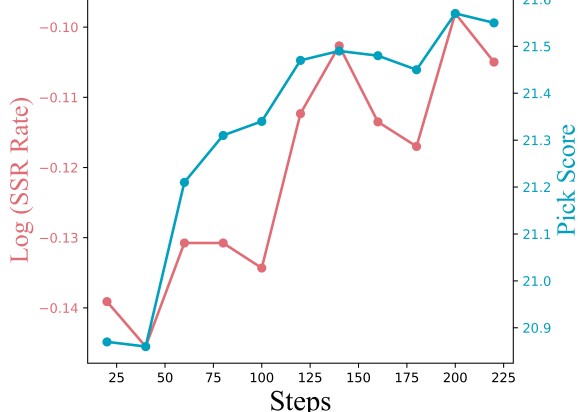

Figure 3: **Correlation between SSR Rate and PickScore.** There is a significant correlation between SSR Rate and PickScore.

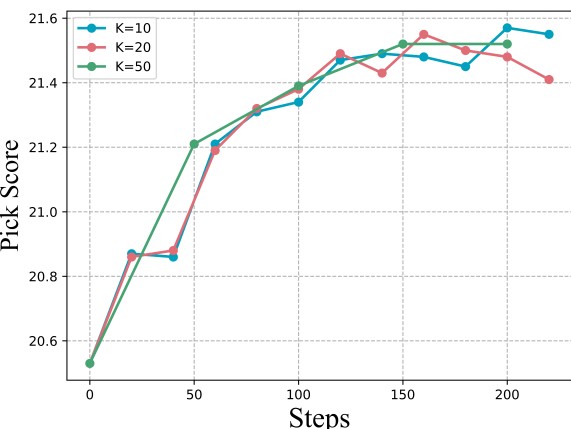

Figure 4: **Ablation study on checkpoint saving frequency.** The checkpoint is saved every k steps. ($K = 10/20/50$)

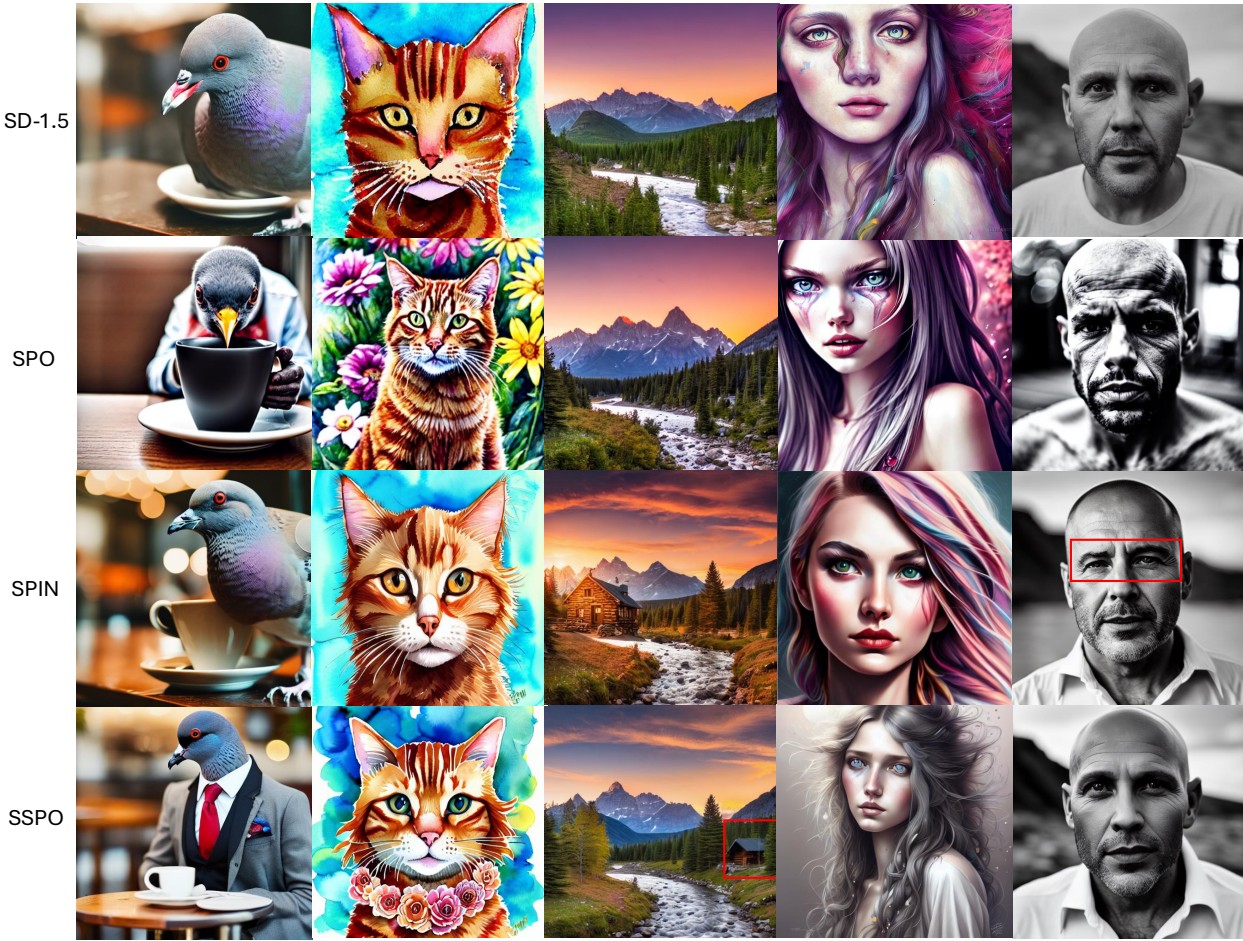

Figure 5: **Text-to-image generation results of SD-1.5, SPO, SPIN-Diffusion and SSPO**. The prompts from left to right are: (1) *Photo of a pigeon in a well tailored suit getting a cup of coffee in a cafe in the morning*; (2) *Ginger Tabby cat watercolor with flowers*; (3) *An image of a peaceful mountain landscape at sunset, with a small cabin nestled in the trees and a winding river in the foreground*; (4) *Detailed Portrait Of A Disheveled Hippie Girl With Bright Gray Eyes By Anna Dittmann, Digital Painting, 120k, Ultra Hd, Hyper Detailed, Complimentary Colors, Wlop, Digital Painting*; (5) *b&w photo of 42 y.o man in white clothes, bald, face, half body, body, high detailed skin, skin pores, coastline, overcast weather.*

Table 4: **Memory & time cost, Extra Hyper-parameters compared to Diff-DPO, and performance comparisons.** Ex Hype. indicates Extra Hyper-parameters compared to Diff-DPO.

| Methods | GPU (MB) | Time (s) | Ex Hype. |
|---------|----------|----------|----------|
| Diff-DPO | **11,206** | **1.36** | **0** |
| D3PO | 17,850 | 18.17 | 3 |
| DPOK | 16,248 | 9.80 | 2 |
| SPIN-Diff | 17,284 | 2.65 | 4 |
| SSPO | 17,284 | 2.65 | 1 |

### 5.2.3   Analysis of Text-to-Video

We further validate SSPO on the text-to-video task in Table 5. A time-lapse video generation task is chosen, and all the models are trained based on AnimateDiff. Our method achieves improvements across all metrics

compared to the vanilla and the SFT AnimateDiff. We visualize video results in Figure 6, showing SSPO achieves higher alignment with the text and better realism compared to the others.

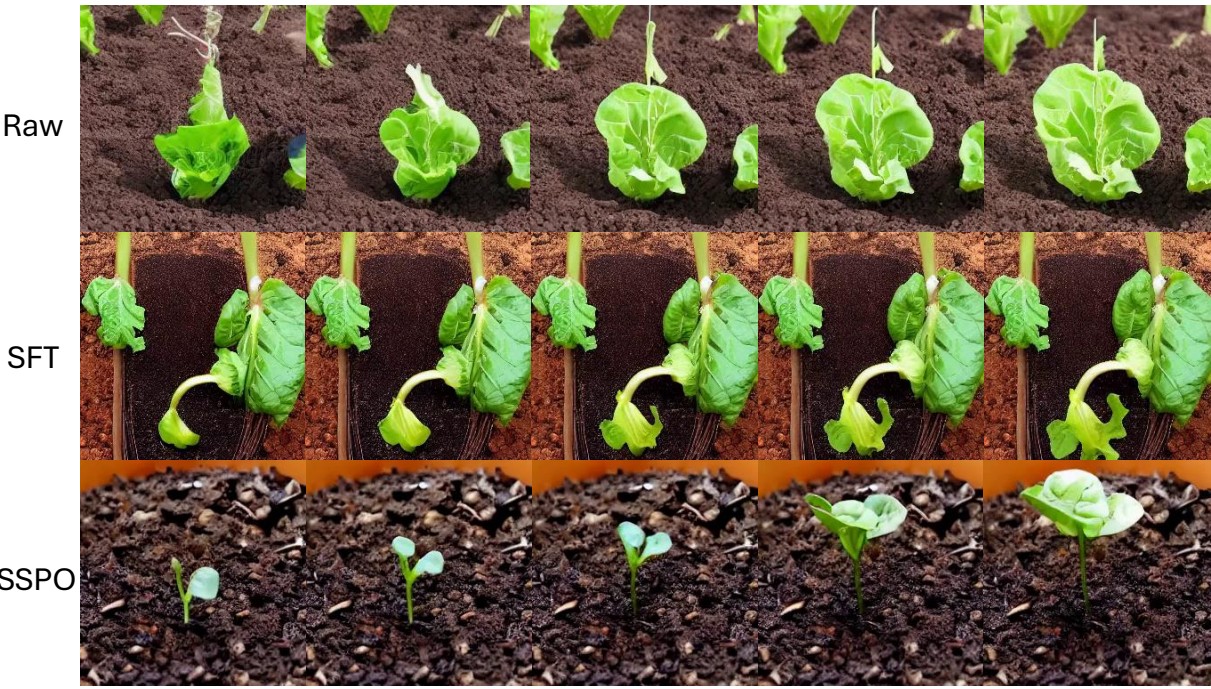

Figure 6: **Text-to-video generation results of AnimateDiff (Raw), SFT, and SSPO.** The prompt is: "*Time-lapse of a lettuce seed germinating and growing into a mature plant. Initially, a seedling emerges from the soil, followed by leaves appearing and growing larger. The plant continues to develop...*"

### 5.2.4 Ablation Study

We perform an ablation study of modules in SSPO in Table 6. When we remove SSR, meaning that all the ERD samples are treated as negative samples, the model's performance drops significantly, demonstrating the effectiveness of SSR. Furthermore, when we replace the sign function $\text{sgn}(x)$ in equation 9 with the indicator function $\mathbb{1}(x > 0)$, *i.e.*, choosing to only learn from the winning image when the ERD image has a higher probability of not being a losing image, and applying DPO when the ERD image has a higher probability of being a losing image, we observe almost no change in performances. This proves that our method is able to filter out "insufficiently negative samples"–samples that are better than the current model's distribution and subsequently boost performance. Furthermore, we examined the necessity of RCR while retaining SSR. When we set the reference model's sampling method to using either the initial checkpoint or the most recent saved checkpoint, we observe a performance drop in both cases. This indicates the effectiveness of our RCR strategy.

We then perform an ablation study w.r.t. the number of iterations $K$. In Figure 4, we found that when changing the total number of iterations $K$ for saving checkpoints, relatively, the larger $K$ achieves better performance. However, the overall trend does not change significantly, which demonstrates the stability of SSPO on $K$.

Table 5: **Metric Scores on the ChronoMagic-Bench-150 Dataset.** ↓ indicates the lower the better, and ↑ indicates the higher the better.

| | FID ↓ | PSNR ↑ | FVD ↓ | MTScore ↑ | CHScore ↑ |
|---|---|---|---|---|---|
| AnimateDiff | 134.86 | 9.18 | 1608.41 | 0.402 | 65.37 |
| SFT | 129.14 | 9.25 | 1415.68 | 0.424 | 70.64 |
| SSPO | **115.32** | **9.36** | **1300.97** | **0.458** | **73.86** |

Table 6: **Ablations on the Pick-a-Pic Validation Dataset.** IR indicates Image Reward.

| Methods | PickScore ↑ | HPSv2 ↑ | IR ↑ | Aesthetic ↑ |
|---|---|---|---|---|
| w/o SSR | 20.88 | 26.78 | 0.366 | 5.491 |
| w/ $\mathbb{1}(x > 0)$ | 21.56 | 27.18 | 0.606 | **5.797** |
| ERD = 0 | 21.41 | 27.04 | 0.562 | 5.727 |
| ERD = $k-1$ | 21.34 | 27.05 | 0.537 | 5.708 |
| SSPO | **21.57** | **27.20** | **0.615** | 5.772 |

## 6 Discussions

### 6.1 Limitations

First, in diffusion models, the theoretical analysis of exploration in policy space constrained by reference models is an open problem. Second, the performance may be further improved if the pixel space (images before encoding) is also considered. We leave this to future work.

### 6.2 Conclusion

In this paper, we propose a Self-Sampling Preference Optimization (SSPO) method that integrates the advantages of SFT and DPO to effectively address the shortcomings of current pre-trained text-to-visual models in aligning with human requirements. We design a Random Checkpoint Replay strategy to construct effective paired data, and both experimental and theoretical analyses have demonstrated its superiority in mitigating overfitting risks and enhancing training stability. Additionally, the proposed Self-Sampling Regularization mechanism dynamically assesses the quality of generated responses to determine whether the samples are truly negative, thereby enabling flexible switching between DPO and SFT and preventing the model from learning biases from non-negative samples. Experimental results show that SSPO achieves significant performance improvements on both text-to-image and text-to-video tasks, validating its effectiveness in improving model generalization. In the future, we plan to further explore more refined negative sample generation strategies and evaluation mechanisms, to achieve even better performance in the broader field of AI-generated content.

### Broader Impact Statement

This work contributes toward scalable and efficient post-training of generative models by reducing the dependence on expensive human annotations or external reward models. In particular, SSPO can democratize model alignment in domains where preference data is scarce, enabling broader adoption of diffusion-based generation techniques in resource-limited research or industry environments. However, the improved accessibility and controllability of powerful generative models also raises concerns. SSPO may inadvertently facilitate the misuse of aligned models in generating misleading or harmful content with higher perceptual quality and human alignment. Moreover, by enabling models to learn preference-aligned behaviors autonomously, there is a risk of encoding and amplifying biases present in the training data without human oversight.

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

# A  Theoretical Analysis

## A.1  Proof of Theorem 4.1 (Generalization)

We first define the generalization loss on the policy as $L(\pi_\theta) := \mathbb{E}_{\mathbf{x} \sim \mathcal{D}} \mathcal{L}(\theta; \mathbf{x})$ and the empirical loss $\widehat{L}(\pi_\theta) := \frac{1}{N} \sum_{i=1}^{N} \mathcal{L}(\theta; \mathbf{x}_i)$, where $N$ is the data size. For the sake of conciseness, we omit the unnecessary superscripts and subscripts in $\mathbf{x}$. We then introduce the PAC-Bayesian theory Germain et al. (2016); Seeger (2002) as our Lemma A.1.

**Lemma A.1** *With a probability larger than $1 - \Delta$, the following equation holds*

$$\mathbb{E}_{\pi_\theta \sim Q} \left[ L(\pi_\theta) \right] \leq \mathbb{E}_{\pi_\theta \sim Q} \left[ \widehat{L}(\pi_\theta) \right] + \sqrt{\frac{2 D_{\mathrm{KL}}(Q || P) + \log \frac{4N}{\Delta^2}}{4N}}, \tag{11}$$

*where $\Delta$ is the confidence radius, $P$ and $Q$ are prior and posterior distributions in the model space, respectively.*

With the maximum entropy, Gaussian initialization is commonly used in policy learning Ahmed et al. (2019); Cen et al. (2022); Glorot & Bengio (2010). Thus, we make the following assumption that aligns well with practical implementations.

**Assumption A.2** *Assume the initial policies obey the Gaussian distribution $\mathcal{N}(0, \sigma_0^2 \mathbf{I})$.*

To simplify the notation in the proof, we shift the indices from $[0, K-1]$ to $[1, K]$, which means the $K$-th checkpoint is the latest one. In the $K$-th iteration, when the reference model is chosen as the last checkpoint, the model parameters are $\theta_K$. When the reference model is chosen uniformly distributed over all previous checkpoints, the expectation of the model parameters are $\frac{1}{K} \sum_{k=i}^{K} \pi_{\theta_k}$. The corresponding distributions are Dirac delta $\delta(\cdot)$ distributions. Thus, we denote the posterior distribution of SPIN $Q_K$ and SSPO $Q_{\mathrm{UNI}}$ as

$$\begin{aligned}
Q_K &= \delta(\pi_\theta - \pi_{\theta_K}), \\
Q_{\mathrm{UNI}} &= \frac{1}{K} \sum_{k=i}^{K} \delta(\pi_\theta - \pi_{\theta_k}).
\end{aligned} \tag{12}$$

Using Lemma A.1, we have the upper bound of the generalization error of SPIN as follows

$$\mathbb{E}_{\pi_\theta \sim Q_K} \left[ L(\pi_\theta) \right] \leq \widehat{L}(\pi_{\theta_K}) + \sqrt{\frac{-2 \log P(\pi_{\theta_K}) + \log \frac{4N}{\Delta^2}}{4N}} := \widehat{L}(\pi_{\theta_K}) + C_K. \tag{13}$$

The upper bound of the generalization error with uniform sampling is

$$\mathbb{E}_{\pi_\theta \sim Q_{\mathrm{UNI}}} \left[ L(\pi_\theta) \right] \leq \frac{1}{K} \sum_{k=i}^{K} \widehat{L}(\pi_{\theta_k}) + \sqrt{\frac{-\frac{2}{K} \sum_{k=1}^{K} \log P(\pi_{\theta_k}) + \log \frac{4N}{\Delta^2}}{4N}} := \frac{1}{K} \sum_{k=i}^{K} \widehat{L}(\pi_{\theta_k}) + C_{\mathrm{UNI}}. \tag{14}$$

We denote the difference $\epsilon$ between the upper bounds as follows

$$\epsilon := \widehat{L}(\pi_{\theta_K}) - \frac{1}{K} \sum_{k=i}^{K} \widehat{L}(\pi_{\theta_k}) + C_K - C_{\mathrm{UNI}}. \tag{15}$$

When $K$ is large, the average loss could be arbitrarily close to the final loss. Thus, the difference term is dominated by the second part $\epsilon_2 := C_K - C_{\mathrm{UNI}}$.

Under Assumption A.2, the logarithmic prior distribution has a closed-form solution. We compare the difference between the KL divergences

$$\epsilon_{\mathrm{KL}} := -\log P(\pi_{\theta_K}) + \frac{1}{K} \sum_{k=1}^{K} \log P(\pi_{\theta_k}) = \frac{\|\pi_{\theta_K}\|^2}{2\sigma_0^2} - \frac{1}{K} \sum_{k=1}^{K} \frac{\|\pi_{\theta_k}\|^2}{2\sigma_0^2}. \tag{16}$$

To calculate the norm of the policies, we trace back to its convergence performance. In SPIN Chen et al. (2024), it shows that $\pi_{\theta_K}$ exponentially convergences to $\pi_{\text{data}}$ (the supervised label with one-hot distribution)

$$\pi_{\theta_K} \propto \pi_{\theta_{K-1}}^{1-\frac{1}{\beta}} \pi_{\text{data}}. \tag{17}$$

Thus, with a large $K$, e.g., when it converges, $\|\pi_{\theta_K}\|^2 = 1$ achieves the upper bound of the second norm in the policy space. Let $\pi_\theta \in \mathbb{R}^d$, where $d$ is the dimension. Then, $\|\pi_{\theta_0}\|^2 = \frac{1}{d}$. This can be easily achieved using the fact that a Gaussian distribution with standard deviation $\sigma$ has the same second norm as a uniform distribution with a radius of $\sigma\sqrt{d}$ in a high dimension. In general, it is a monotonic process. Without loss to generality, we simplify the convergence process as a linear process to show the result. Thus, we have

$$\epsilon_{\text{KL}} = \frac{1}{2\sigma_0^2}(1 - \frac{1+\frac{1}{d}}{2}) = \frac{1}{4\sigma_0^2}(1 - \frac{1}{d}) > 0. \tag{18}$$

Finally, we have

$$\epsilon_2 = \frac{\epsilon_{\text{KL}}}{2N(C_K + C_{\text{UNI}})} > 0, \tag{19}$$

which shows the uniformly sampling method has a lower confidence upper bound (The loss is smaller.) compared to the latest checkpoint selection method. Thus, the uniformly sampling method achieves better generalization ability with less overfitting risk.

## A.2 Proof of Theorem 4.2

In step $t$, recall that the original image is $\mathbf{x}_0 = \frac{\mathbf{x}_t - \sqrt{1-\alpha^t}\epsilon}{\sqrt{\alpha^t}}$, and the image recovered by the DDPM $\theta$ is defined as $\widehat{\mathbf{x}}_0^\theta := \frac{\mathbf{x}_t - \sqrt{1-\alpha^t}\epsilon_\theta(\mathbf{x}_t)}{\sqrt{\alpha^t}}$. We denote the means of these two Gaussian distributions as $\widehat{\mu}_0^\theta$ and $\mu_0$, respectfully. The performance of a DDPM model $\theta$ can be measured by the KL distance (Chan, 2024) $D_{\text{KL}}(\widehat{\mathbf{x}}_0^\theta||\mathbf{x}_0)$ between the recovered image $\widehat{\mathbf{x}}_0^\theta$ and the original image $\mathbf{x}_0$. Thus, we give the standard of recovery performance from noisy image $\mathbf{x}_t$ in Definition A.3.

**Definition A.3** *Given two DDPMs $\theta_1$ and $\theta_2$, $\theta_1$ is better than $\theta_2$ ($\theta_1 \succ \theta_2$) if its predicted image $\widehat{\mathbf{x}}_0^{\theta_1}$ has less KL divergence with the original image $\mathbf{x}_0$ as follows*

$$D_{\text{KL}}(\widehat{\mathbf{x}}_0^{\theta_1}||\mathbf{x}_0) \leq D_{\text{KL}}(\widehat{\mathbf{x}}_0^{\theta_2}||\mathbf{x}_0). \tag{20}$$

In the noise injection process, the variance of the samples remains the same. ($\mathbf{x}_0$ and $\mathbf{x}_t$ have the same variance.) With the fact that the KL divergence between two Gaussian distributions with the identical variance is proportional to the Euclidean distance of their means, we have

$$
\begin{aligned}
D_{\text{KL}}(\widehat{\mathbf{x}}_0^\theta||\mathbf{x}_0) &= \frac{1}{2\sigma_0^2}\|\widehat{\mu}_0^\theta - \mu_0\|^2 \\
&= \frac{1}{2\sigma_0^2}\left\|\mathbb{E}\left[\frac{\mathbf{x}_t - \sqrt{1-\alpha^t}\epsilon_\theta(\mathbf{x}_t)}{\sqrt{\alpha^t}}\right] - \mathbb{E}\left[\frac{\mathbf{x}_t - \sqrt{1-\alpha^t}\epsilon}{\sqrt{\alpha^t}}\right]\right\|^2 \\
&= \frac{1-\alpha^t}{2\sigma_0^2\alpha^t}\left\|\mathbb{E}\left[\epsilon_\theta(\mathbf{x}_t)\right] - \mathbb{E}[\epsilon]\right\|^2 \\
&= \frac{1-\alpha^t}{2\sigma_0^2\alpha^t}\mathbb{E}\left[\|\epsilon_\theta(\mathbf{x}_t) - \epsilon\|^2\right] - \frac{1-\alpha^t}{2\sigma_0^2\alpha^t}.
\end{aligned}
\tag{21}
$$

The last step is from Jensen's inequality for the square-error function. Thus, given the condition

$$\mathbb{E}\left[\|\epsilon_{\theta_1}(\mathbf{x}_t) - \epsilon\|^2 - \|\epsilon_{\theta_2}(\mathbf{x}_t) - \epsilon\|^2\right] \leq 0, \tag{22}$$

we have

$$D_{\text{KL}}(\widehat{\mathbf{x}}_0^{\theta_1}||\mathbf{x}_0) - D_{\text{KL}}(\widehat{\mathbf{x}}_0^{\theta_2}||\mathbf{x}_0) \leq 0. \tag{23}$$

The prediction from model $\theta_1$ has a smaller KL distance compared to the prediction from model $\theta_2$. Thus, $\theta_1$ recovers better samples and $\theta_1 \succ \theta_2$ by the definition of performance measurement. As a result, $\theta_1$ has a higher probability of generating a good result $\mathbf{x}_0^{\theta_1}$.

# B Supplementary Experiments

## B.1 Results on the Parti-prompt Dataset.

From the results in Table. 7, it is evident that SSPO achieves the highest scores on key metrics such as PickScore and HPSv2, demonstrating its superior overall performance in meeting user preferences and generating visual content, even comparable to LS-Diff Zhang et al. (2024) which utilizes SDv2 as the base model. Meanwhile, SSPO maintains a high level of image quality (Aesthetic) and a stable image reward (IR), while still outperforming other methods in overall evaluations, which highlights its distinct advantage in balancing generation quality and alignment with human needs.

Table 7: **Model Feedback Results on the Parti-prompt Dataset.** IR indicates Image Reward. * indicates that the base model of LS-Diff is SDv2.

| Methods | PickScore ↑ | HPSv2 ↑ | IR ↑ | Aesthetic ↑ |
|---------|-------------|---------|------|-------------|
| SD-1.5 | 21.38 | 26.70 | 0.16 | 5.33 |
| SFT | 21.68 | 27.40 | 0.56 | 5.53 |
| Diff-DPO | 21.63 | 26.93 | 0.32 | 5.41 |
| Diff-RPO | 21.66 | 27.05 | 0.39 | 5.43 |
| SPO | 21.85 | 27.41 | 0.42 | 5.63 |
| SPIN-Diff | 21.91 | 27.58 | 0.51 | **5.78** |
| LS-Diff* | - | - | 0.73 | 5.48 |
| SSPO | **21.93** | **27.79** | **0.58** | 5.64 |

## B.2 Reward Distribution

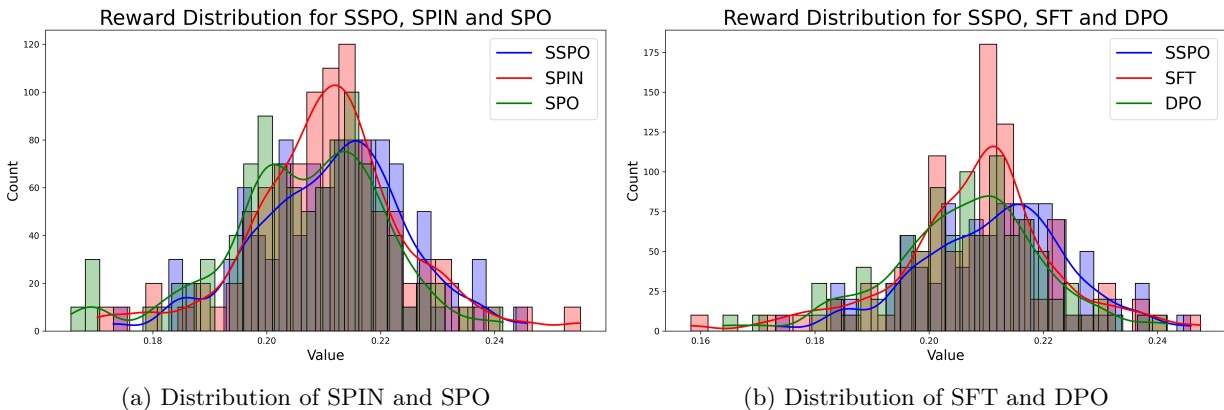

(a) Distribution of SPIN and SPO          (b) Distribution of SFT and DPO

Figure 7: Reward distribution of several methods.

From the comparison of the reward distributions of different methods in the Figure. 7, it is evident that SSPO, compared to other methods (e.g., SPIN, SPO, SFT, DPO), demonstrates the following significant advantages:

**Lower kurtosis in the reward distribution:** As shown by the histogram and corresponding fitted curves, the SSPO distribution curve is flatter, and its peak is less pronounced than that of other methods. This indicates that SSPO's reward distribution is more uniform during training, with fewer extremely high or low values, leading to better training stability.

**Higher overall reward level:** The bar chart illustrates that both the mean and median rewards obtained by SSPO are relatively higher, and the distribution is shifted to the right. This suggests that under the same training environment and task setup, SSPO can achieve better returns in most scenarios.

**More balanced distribution:** While other methods may produce extremely high or low rewards in certain cases, SSPO's reward distribution is more concentrated. It not only attains a higher average reward but also effectively avoids excessive dispersion, thus enhancing overall robustness.

In summary, the reward distribution comparison in the figure indicates that SSPO, through more stable and efficient policy learning, achieves higher overall returns while maintaining a smoother distribution. It exhibits a distinctly comprehensive advantage across various metrics.

## B.3 Human Evaluation

We conduct a human evaluation experiment based on the Pick-a-Pic validation dataset. Following D3PO (Yang et al., 2024), each image in the dataset was assessed by 5 human raters. We then report the average percentage of images that received favorable evaluations in Figure. 8. The results demonstrate the superiority of SSPO compared to SPIN-diffusion.

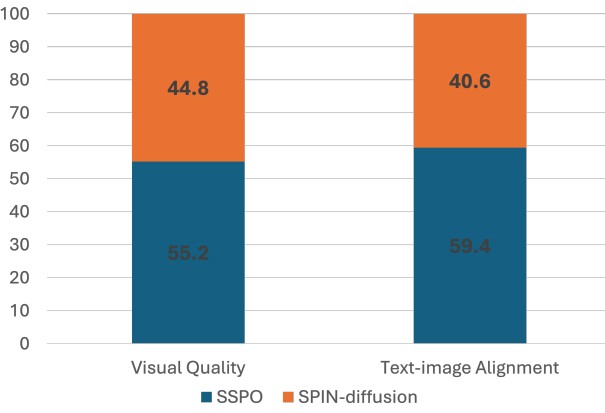

Figure 8: Human evaluation result of SSPO and SPIN-diffusion

## B.4 Step-wise Performance for SFT and SSPO

We then display the changes in PickScore during the training process of SSPO and SFT in Figure 9. SFT quickly converges and begins to fluctuate, while SSPO is able to steadily improve throughout the training process.

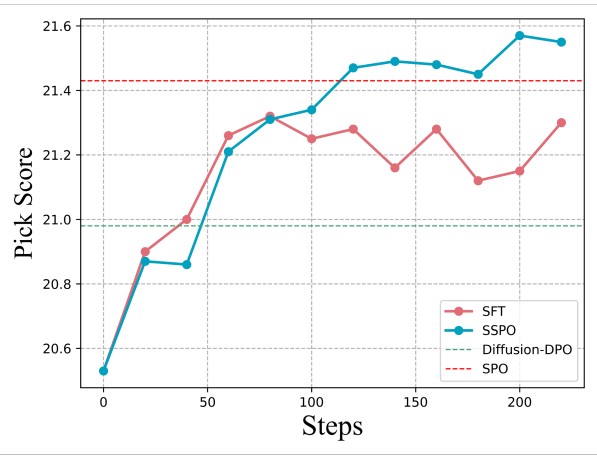

Figure 9: Step-aware performance for SFT and SSPO.

### B.5 Pseudocode

In this subsection, we give the pseudocode of the loss function for training.

```python
def loss(target, model_pred, ref_pred, scale_term):
    #### START LOSS COMPUTATION ####
    if args.train_method == 'sft': # SFT, casting for F.mse_loss
        loss = F.mse_loss(model_pred.float(), target.float(),
                          reduction="mean")
    elif args.train_method == 'SSPO':
        # model_pred and ref_pred will be
        # (2 * LBS) x 4 x latent_spatial_dim x latent_spatial_dim
        # losses are both 2 * LBS
        # 1st half of tensors is preferred (y_w),
        # second half is unpreferred

        model_losses = (model_pred - target).pow(2).mean(dim=[1,2,3])
        model_losses_w, model_losses_l = model_losses.chunk(2)

        with torch.no_grad():
        # Get the reference policy (unet) prediction
            ref_pred = ref_unet(*model_batch_args,
                                added_cond_kwargs=added_cond_kwargs
                                ).sample.detach()

            ref_losses = (ref_pred - target).pow(2).mean(dim=[1,2,3])
            ref_losses_w, ref_losses_l = ref_losses.chunk(2)

        pos_losses_w = model_losses_w - ref_losses_w

        sign = torch.where(pos_losses_w > 0,
                              torch.tensor(1.0),
                              torch.tensor(-1.0))

        model_diff = model_losses_w + sign * model_losses_l
        ref_diff = ref_losses_w + sign * ref_losses_l

        scale_term = -0.5 * args.beta_dpo
        inside_term = scale_term * (model_diff - ref_diff)
        implicit_acc = (inside_term > 0).sum().float() /
                        inside_term.size(0)
        loss = -1 * (F.logsigmoid(inside_term)).mean()

    return loss
```

### B.6 Visual Generation Examples

We present more text-to-visual generation results of SSPO and other methods. In Figure 10, we show the text-to-image generation results of SD-1.5, SPO, SPIN-Diffusion, and SSPO. In Figure 11 and Figure 12, we show the text-to-video generation results of AnimateDiff (Raw), SFT, and SSPO.

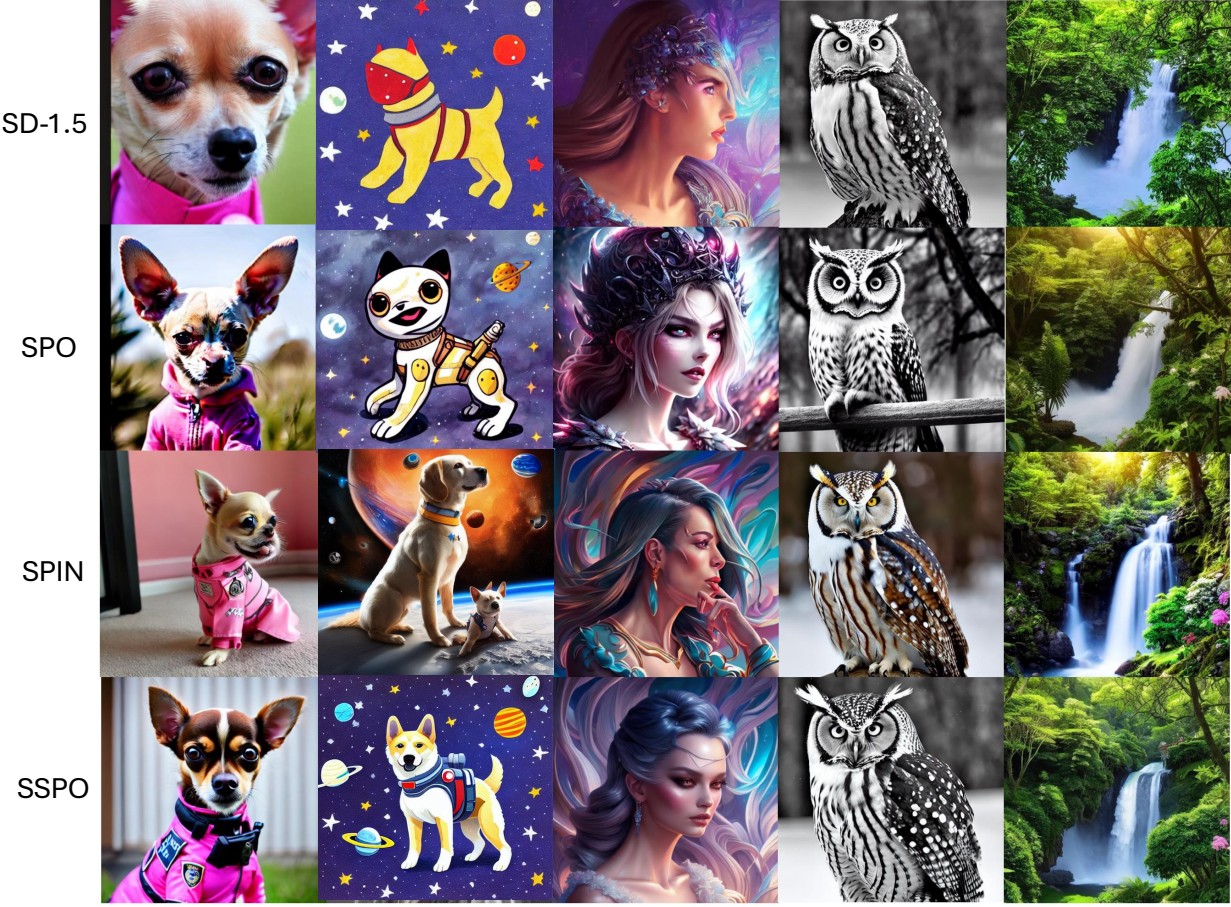

Figure 10: **Text-to-image generation results of SD-1.5, SPO, SPIN-Diffusion, and SSPO.** Prompts from left to right: (1) *Pink Chihuahua in police suit*; (2) *Space Dog*; (3) *Chic Fantasy Compositions, Ultra Detailed Artistic, Midnight Aura, Night Sky, Dreamy, Glowing, Glamour, Glimmer, Shadows, Oil On Canvas, Brush Strokes, Smooth, Ultra High Definition, 8k, Unreal Engine 5, Ultra Sharp Focus, Art By magali villeneuve, rossdraws, Intricate Artwork Masterpiece, Matte Painting Movie Poster*; (4) *winter owl black and white*; (5) *You are standing at the foot of a lush green hill that stretches up towards the sky. As you look up, you notice a beautiful house perched at the very top, surrounded by vibrant flowers and towering trees. The sun is shining brightly, casting a warm glow over the entire landscape. You can hear the sound of a nearby waterfall and the gentle rustling of leaves as a gentle breeze passes through the trees. The sky is a deep shade of blue, with a few fluffy clouds drifting lazily overhead. As you take in the breathtaking scenery, you can't help but feel a sense of peace and serenity wash over you.*

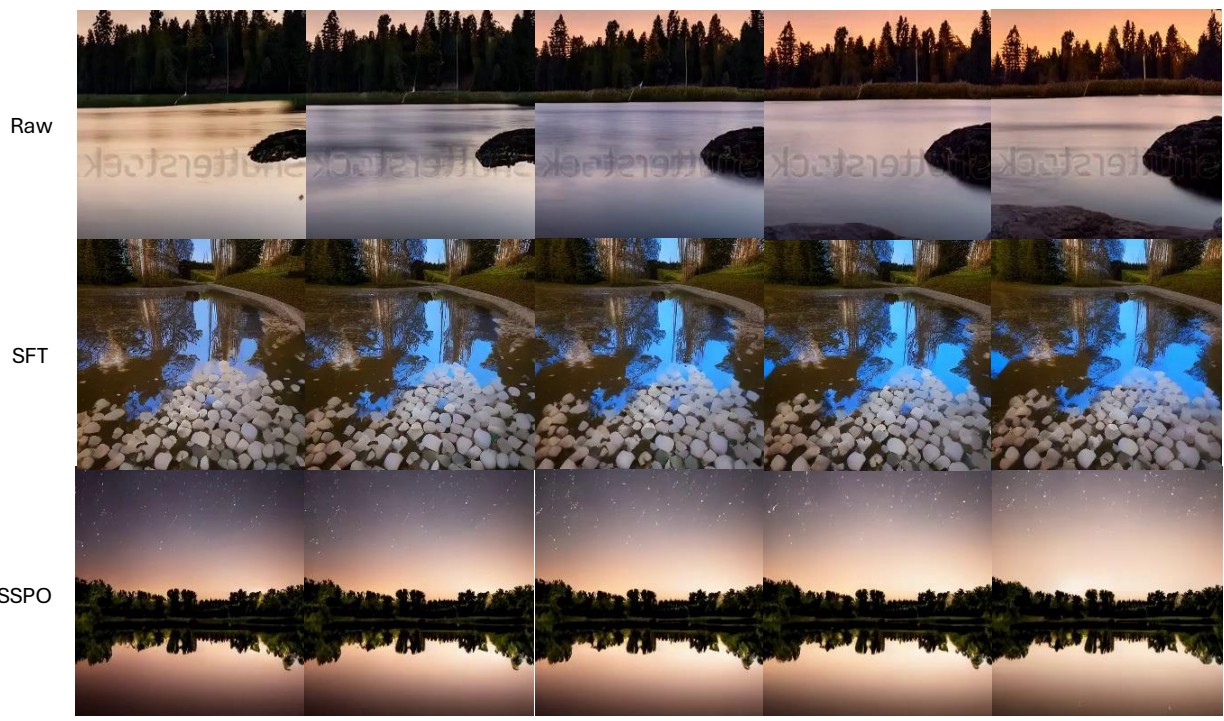

Figure 11: **Text-to-video generation results of AnimateDiff (Raw), SFT, and SSPO.** Prompt: "*Time-lapse of night transitioning to dawn over a serene landscape with a reflective water body. It begins with a starry night sky and minimal light on the horizon, progressing through increasing light and a glowing horizon, culminating in a serene early morning with a bright sky, faint stars, and clear reflections in the water.*"

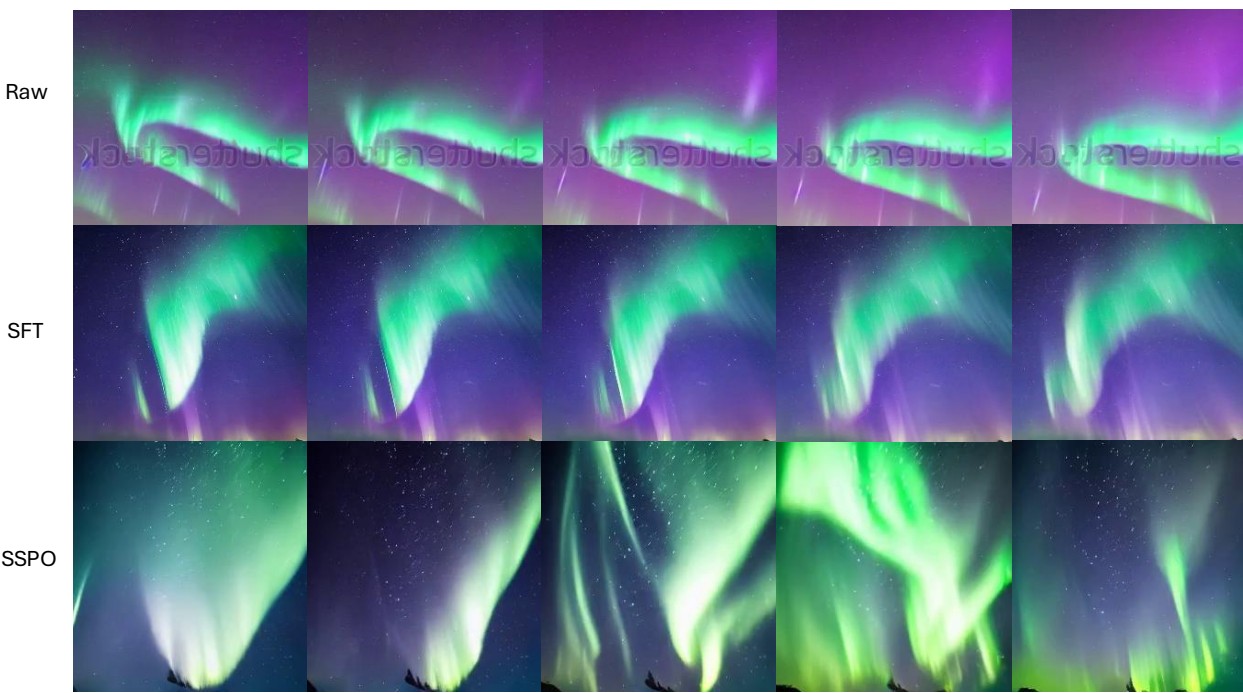

Figure 12: **Text-to-video generation results of AnimateDiff (Raw), SFT, and SSPO.** Prompt: "*Time-lapse of aurora borealis over a night sky: starting with green arcs, intensifying with pronounced streaks, and evolving into swirling patterns. The aurora peaks in vivid hues before gradually fading into a homogeneous glow on a steadily brightened horizon.*"

