# OpenReview forum: "Bridging SFT and DPO for Diffusion Model Alignment with Self-Sampling Preference Optimization"
_TMLR — Rejected by TMLR_

### Review · Reviewer_HgGi · 2025-08-02

**Summary Of Contributions:**

The paper proposed a new alignment method for diffusion models called Self-Sampling Preference Optimization (SSPO). It includes a Random Checkpoint Replay
(RCR) strategy by randomly sampling from previous checkpoints, instead of the earliest or latest one, and a Self-Sampling Regularization (SSR) strategy that incorporates a sign (+1 or -1) into the loss. Experiments are conducted on both image and video generation.

**Audience:**

Yes

**Audience Explanation:**

The proposed algorithm is built upon existing diffusion model alignment methods such as diffusion-DPO or SPIN-diffusion. It conducts an ablation study and experiments to support its claim.

**Broader Impact Concerns:**

It has been addressed in the Broader Impact Statement section.

**Claims And Evidence:**

No

**Claims Explanation:**

After reading the paper, I find some parts unclear.

1. The paper studies the alignment of diffusion models. However, the abstract does not mention the diffusion model (or text-to-image models) until the end. Moreover, RL, SFT, and DPO are originally alignment methods for language models. The description of the work is ambiguous.

2. The background section makes some claims that are very strange. “To maximize the evidence lower bound (ELBO), we usually minimize the loss function.” In my understanding, ELBO is usually used in the theory of VAE, while the diffusion model trains an estimated score function. The connection between them is not straight. “T as a pre-specified constant in is ignored in the loss function, because it has no contribution to training.” T is used to define the forward process; thus, this claim is not correct.

3. Introducing DPO in the background part is not enough for understanding. In fact, it should discuss algorithms on diffusion models like, diffusion DPO, SPIN-diffusion,..., not just the original DPO for language model training. This leads to confusion when reading. For example, equation (5) would be exactly the same as the loss for diffusion DPO, except the sampling distribution. This point is not made clear in the paper.

4. The proven theory is not rigorous. It does not help to explain why the proposed method is meaningful.

Theorem 4.1: The result can only compare two theoretical upper bounds, instead of the loss. The big O notation is also weird.

Theorem 4.2: “generates better prediction” is not well defined. Equation (21), the last equation should be an inequality, then the results following do not hold.

5. Some other issues: reference modelâĂŹs (Page 2), Page 5: SPIN should be SPIN-diffusion, DITTO (no reference), SPPO (it should be SSPO)

**Requested Changes:**

The paper should emphasize that the proposed method is specifically designed for diffusion models, not language models, and ensure that all the baselines are also designed for diffusion models.

---

> ### Author Response · Authors · 2025-11-11
>
> 1. Thank you for the suggestion. We rewrite the abstract to introduce the diffusion model earlier as suggested.
>
> 2. We follow the same standard process in Diffusion-DPO [1] and this diffusion tutorial [2]. ELBO is widely used in diffusion models. To avoid the misunderstanding of $T$, we have deleted the sentence about $T$.
>
> 3. Thanks for the suggestion. Considering the length requirements, we refer to the previously proposed method in related works. We aim to introduce the background, and use the previous results (and follow the same standard process) in the above-mentioned diffusion DPO papers, e.g., Diffusion-DPO and SPIN-Diffusion. (There is a typo in the Diffusion-DPO paper. It should maximize the ELBO, instead of minimizing it.)
>
> 4. (1) Yes. It is used to compare the upper bounds, for which the big O notation is designed. "A description of a function in terms of big O notation only provides an upper bound on the growth rate of the function." https://en.wikipedia.org/wiki/Big_O_notation
>
> (2) Thanks for bringing this. We add extra justification to enhance the readability. We also briefly explain it here: We have $tr(Cov(\epsilon_{\theta} - \epsilon)) = d$, and it is an equality. We have unit variance noise $E[||\epsilon||^{2}] = 1$, and the model prediction has the same marginal variance as the true noise $E[||\epsilon_{\theta}||^{2}] = 1$. We know that the model’s predicted noise and the actual Gaussian noise are independent (or at least orthogonal in expectation), $E[\epsilon_{\theta}^{T} \epsilon ] = 0$. Thus, we have $tr(Cov(\epsilon_{\theta} - \epsilon)) = d$. The loss is averaged per dimension (which they typically do in DDPM derivations), and $tr(\cdot) / d = 1$.
>
> 5. Thanks for pointing them out. We have fixed the typos about the reference.
>
> [1] Diffusion Model Alignment Using Direct Preference Optimization. CVPR 2024.
>
> [2] Tutorial on Diffusion Models for Imaging and Vision. arXiv:2403.18103, 2024.

---

### Review · Reviewer_mmys · 2025-09-01

**Summary Of Contributions:**

Strengths.
1. Simple and practical design that combines SFT’s training stability with DPO-like generalization, without requiring an RM or paired preference data.
2. RCR improves stability and generalization via uniform sampling over past checkpoints, with PAC-Bayesian support.
3. SSR uses a winner-anchored criterion for per-sample switching, avoiding penalizing high-quality samples and strengthening effective gradients.
4. Strong results across both image (SD-1.5/SDXL) and video (AnimateDiff) settings and multiple OOD benchmarks; small resource/hyperparameter overhead.

Weaknesses.
1. Frequent checkpointing can significantly affect training efficiency; when scaling from single-GPU to multi-GPU/multi-node under similar CPU I/O bandwidth, efficiency degrades noticeably.
2. Hyperparameter and implementation sensitivity are under-explored, especially the choice of checkpoint save frequency.

**Audience:**

Yes

**Audience Explanation:**

Yes.
SSPO offers a practical, Reward-Model-free alternative that bridges SFT and DPO, with solid gains on text-to-image and text-to-video diffusion models. Its simple replay-and-switching mechanism, theoretical backing, and strong empirical results on standard and OOD benchmarks would interest practitioners seeking stable, scalable alignment methods without costly preference data or reward models.

**Claims And Evidence:**

Yes

**Claims Explanation:**

Yes.
1. The paper reports consistent gains on image (SD-1.5/SDXL) and video (AnimateDiff) across multiple benchmarks (Pick-a-Pic val, HPSv2, Parti-Prompts; FID/FVD/MT/CH for video).
2. Clear ablations isolate RCR and SSR. Removing SSR or changing the replay policy (only initial or only latest checkpoint) degrades performance, supporting the necessity of both components.
3. A PAC-Bayesian generalization argument for uniform replay and a sufficient-condition result linking winner-anchored noise loss to sample polarity provide plausible rationale.

**Requested Changes:**

1. Provide scalability and efficiency profiling: end-to-end throughput and wall-clock comparisons vs. key baselines under single-GPU, multi-GPU  settings, including CPU/I/O bottlenecks from checkpoint replay.

2.Expand hyperparameter robustness, systematic sweeps for checkpoint save frequency.

---

> ### Author Response · Authors · 2025-11-11
>
> **Efficiency.**
> Thanks for the suggestion. We report the GPU memory footprint and per-iteration wall-clock time in Table 4. Under identical hardware and batch settings, SSPO uses less memory and runs faster per iteration than most competing methods, indicating that RCR + SSR adds negligible overhead beyond DPO-like training. In practice, we set the checkpoint save frequency (K = 20 steps), and the method typically peaks around 200 steps, so only about ten checkpoints are materialized before early stopping. This keeps checkpointing I/O amortized and the end-to-end efficiency impact minimal.
>
> **Hyperparameter robustness.**
> Compared to Diffusion-DPO, SSPO introduces only one additional hyperparameter—the checkpoint save frequency K. We present a systematic ablation in Figure 4, which shows that varying K has only a minor effect on PickScore, demonstrating the stability of our method over a reasonable range of settings.

---

### Review · Reviewer_Xuq5 · 2025-10-27

**Summary Of Contributions:**

This paper addresses the problem of aligning text-to-visual diffusion models with human preferences, aiming to combine the stability of Supervised Fine-Tuning (SFT) with the generalization capabilities of preference-based methods like DPO, but without requiring explicit preference pairs or a reward model (RM).

**Strengths:**

1. This is an important problem statement as it deals with the fundamental generation process. Any improvement here will have an impact on the downstream processes.

2. The paper is clearly written and easy to understand.

3. The ablation studies in Table 6 are clear and effective. They successfully demonstrate that both RCR (by ablating to ERD=0 and ERD=k-1) and SSR (by ablating w/o SSR) are critical components that contribute to the method's final performance.

4. The generalization proof is important and provides theoretical backing to the use of random checkpoint sampling.

**Weakness:**

1. I am not able to understand why the model performance peaks in Figure 2 and then decreases for all the methods. Can you explain this?
2. Most of the results (Tables 1, 2, 3) have very small margins compared to the second-best result. In this light, it is hard to comment on the performance improvement of the method.
3. Although model generalization was proved in theory, there are experiement supporting them are not strong.

**Audience:**

Yes

**Audience Explanation:**

The paper deals with optimization in a text-to-visual diffusion model, a large section of TMLR's audience will be interested in its findings.

**Broader Impact Concerns:**

There is a Broader Impact Section that accurately describes the impact of the work.

**Claims And Evidence:**

No

**Claims Explanation:**

The most interesting claim to me in this work was the generalization claim. It was proved theoretically with some major assumptions, specifically, "Without loss of generality, we simplify the convergence process as a linear process to show the result". I am not sure how the convergence process is considered to be linear.

Similarly, the experiment supporting this claim is also not strong enough for me.

**Requested Changes:**

1) Explain why the model performance first rises and then it overfits on the data.

2) For all experiments, add standard deviation or standard error to the final calculated score with at least 3 different seeds for each method. This is very much needed as the final scores are very close and, in my opinion, within the margin of error for all the methods.

3) The assumptions needed for the proof of generalization need to be more clearly stated.

---

> ### Author Response · Authors · 2025-11-11
>
> On performance margins for SSPO:
>
> While the absolute margins may appear small (for example, +0.62 PickScore and +0.615 Image Reward on Pick-a-Pic; +0.70 IR on HPSv2; +1.16 IR and +0.38 PickScore on SDXL), they are consistent across all datasets and backbones (SD-1.5, SDXL, AnimateDiff), indicating robust and repeatable gains rather than noise.
> Importantly, our training setup is more constrained than most preference-based baselines:
>
> 1. SSPO uses only the SFT-style single-sample data, without any paired preference annotations or reward model supervision, whereas DPO, Diff-DPO, and SPIN-Diff rely on paired positive/negative samples or explicit preference rewards.
>
> 2. This means that our model achieves comparable or superior performance under strictly weaker supervision.
>
> From a fairness standpoint, this makes SSPO’s improvements nontrivial—it bridges the gap between pure SFT (stable but less aligned) and preference-based methods (data-hungry and costly) with a lightweight, reward-model-free procedure.
>
> ---
>
> Model performance fluctuation:
>
> The rise–then–fall pattern is expected under noisy-preference benchmarks, and the principal contributing factor is dataset quality. The Pick-a-Pic benchmark contains a substantial portion of prompts/images sourced from Dreamlike Photoreal 2.0 (PickScore ≈ 20.7; https://arxiv.org/pdf/2307.10159) and Stable Diffusion 2.1 (PickScore ≈ 20.3; https://openreview.net/pdf?id=ruZksIJBBd), which imposes a relatively low performance ceiling and high heterogeneity. As training progresses, models initially exploit the reliable signal and subsequently overfit to annotation/model noise, yielding a peak followed by regression. A comparable trajectory is reported in SPIN-diffusion (Figures 6–7).

---

### Decision · Action_Editor_c7ve · 2026-01-13

**Recommendation:** Reject

**Audience:**

Yes

**Audience Explanation:**

This paper addresses the task of aligning diffusion models for text-to-image and text-to-video tasks,  which will be of interest to several members in TMLR's audience.

**Claims And Evidence:**

No

**Claims Explanation:**

This paper addresses the challenging task of aligning diffusion models for text-to-image and text-to-video tasks. Since supervised fine-tuning and reinforcement learning both have their advantages and limitations, this paper proposes a new alignment method, Self-Sampling Preference Optimization (SSPO). One of the highlights of the paper is the Random Checkpoint Replay which utilizes historical checkpoints for obtaining paired data, which mitigates overfitting. In addition, no explicit preference pairs are required.

The paper was reviewed by three reviewers and after the discussion stage, two reviewers were not fully satisfied that the claims of the paper are fully supported by the experiments. Specifically, the claims regarding model generalization, the rise and fall in performance,etc. are not satisfactorily justified by the experiments,  At this stage, the paper requires more work to reduce the ambiguity of the claims and to fully justify them using the experimental results. This is the primary reason for the decision.

In addition, the reviewers also mention the limited novelty of the work, which further supports the decision, but has not been considered to arrive at the decision for the paper.

**Resubmission Of Major Revision:**

The authors may consider submitting a major revision at a later time.